# Station-Keeping Control of Autonomous and Remotely-Operated Vehicles for Free Floating Manipulation

Ningning Ding [1,2,3,4], Yuangui Tang [1,2,3,*], Zhibin Jiang [1,2,3], Yunfei Bai [1,2,3,4] and Shixun Liang [1,2,3,4]

1   State Key Laboratory of Robotics, Shenyang Institute of Automation, Chinese Academy of Sciences, Shenyang 110016, China; dingnn@mail.ustc.edu.cn (N.D.); jiangzhibin@sia.cn (Z.J.); baiyunfei@sia.cn (Y.B.); liangshixun@sia.cn (S.L.)
2   Institutes for Robotics and Intelligent Manufacturing, Chinese Academy of Sciences, Shenyang 110169, China
3   Key Laboratory of Marine Robotics, Shenyang 110169, China
4   University of Chinese Academy of Sciences, Beijing 100049, China
*   Correspondence: tyg@sia.cn

**Abstract:** This paper investigates the station-keeping control of autonomous and remotely-operated vehicles (ARVs) for free-floating manipulation under model uncertainties and external disturbances. A modified adaptive generalized super-twisting algorithm (AGSTA) enhanced by adaptive tracking differentiator (ATD) and reduced-order extended state observer (RESO) is proposed. The ATD is used to obtain the smooth reference signal and its derivative. The RESO is used to estimate and compensate for the model uncertainties and external disturbances in real-time, which enhances the robustness of the controller. The modified AGSTA ensures the fast convergence of the system states and maintains them in a predefined neighborhood of origin without overestimating control gains. Besides, the proposed new variable gain strategy completely avoids the control gains vibrating near the set minimum value. Thanks to the RESO, the proposed controller is model-free and can be easily implemented in practice. The stability of the closed-loop system is analyzed based on Lyapunov's direct method in the time domain. Finally, the proposed control scheme is applied to the station-keeping control of Haidou-1 ARV, and the simulation results confirm the superiority of the proposed control scheme over the original AGSTA.

**Keywords:** station-keeping control; underwater vehicle-manipulator system; super-twisting algorithm; extended state observer

## 1. Introduction

In recent years, unmanned underwater vehicles (UUVs) have been widely utilized in various areas such as marine science, marine rescue, and offshore industry [1]. A considerable number of these applications require UUVs have intervention capabilities. Currently, most intervention tasks are faced up by a remotely operated vehicle (ROV) equipped with one or multiple manipulators. However, the ROV needs to be controlled by highly skilled operators via a master-slave approach, which increases human fatigue over time and has a more significant time delay in the control loop [2]. Furthermore, the ROV requires an expensive support vessel equipped with dynamic positioning (DP) systems and capable of handling the umbilical cable. As a result, the ROV can only work in a small zone due to the umbilical cable's restriction. To overcome these limitations, ARV has been proposed for deep-sea exploration and intervention. The ARV communicates with the support vessel through an optical fiber instead of an umbilical cable. Thus, the ARV can deploy and operate from vessels lacking DP capabilities, which reduces operational costs [3]. The main features of the ARV lie in that it can perform large-scale exploration tasks, monitoring tasks, and underwater intervention tasks in different operational modes.

The ARV we investigate in this paper is called Haidou-1. It is composed of a fully-actuated vehicle and a six DoFs deep-sea electric manipulator, as shown in Figure 1. It

should note that Haidou-1 is a typical underwater vehicle manipulator system (UVMS). Thus, the research on UVMS can also be applied to our research. When the Haidou-1 performs free-floating manipulation, such as seabed sampling, it will be desired that the vehicle has a station-keeping function (the vehicle's ability to maintain the same position and attitude at all times even under disturbances). However, in order to achieve stability and maneuverability simultaneously, the metacentric height of the ARV is set lower than that of a conventional ROV. Besides, the ARV thrusters' performance is weaker than the traditional ROV. Therefore, the dynamic coupling effect caused by the motion of the manipulator has a substantial impact on the vehicle. The vehicle will deviate from the desired position and attitude, thereby severely reducing the accuracy of the manipulator's end-effector. Therefore, the station-keeping control of the vehicle is essential for free-floating manipulation. However, the design of the controller is quite challenging work due to the coupling effects, strong nonlinearity, random external disturbances, unpredicted ocean currents, and the difficulty in accurately modeling the hydrodynamic effects [4].

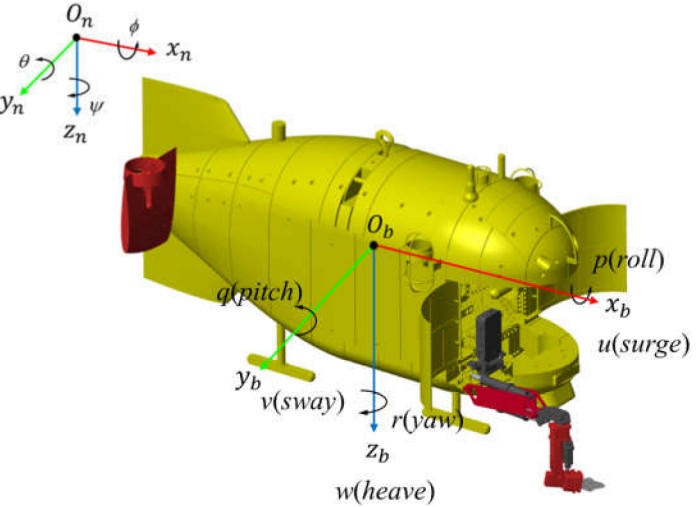

**Figure 1.** The Haidou-1 ARV and its reference frames.

There exist two main strategies for UVMS control in the literature. The first strategy controls the whole system at the operational space level, treating the UVMS as a single system [5–7]. Another strategy aims to decouple the UVMS and independently control the vehicle and the manipulator [8]. When the vehicle and manipulator have similar dynamic characteristics or bandwidth, the first strategy is feasible. However, Haidou-1 has a big difference in vehicle and manipulator inertias. Moreover, the UVMS as a whole is a kinematically redundant system that is hard to handle. Thus, we choose the decoupled control strategy in this paper. Specifically, the vehicle performs station-keeping control to keep it in a fixed attitude and position, and the manipulator carries out the prescribed tasks alone. The coupling force caused by manipulator movement is considered as a disturbance to be compensated by the vehicle controller. In recent years, this approach has been investigated by several researchers.

Koval [9] proposed an automatic stabilization method for the vehicle where the manipulator disturbance term is compensated by manipulator kinematics and simplified dynamics. McLain et al. [10] proposed a model-based coordinated control strategy. Coupling reaction forces duo to manipulator motion were predicted and counteracted by the vehicle controller. Antonelli and Cataldi [11] designed a recursive and adaptive controller for the vehicle to mitigate the dynamic interaction caused by the manipulator. An alternative vehicle stabilization strategy is proposed in [12], where the focus is to control the UVMS barycenter. This strategy has a significant reduction in power consumption compared to the feedback and feedforward control. An observer-based controller has been proposed in [13], where the disturbance force observer is used to estimate the coupling

and restoring forces induced by the operating manipulator. Besides, the restoring force is reduced through a motion planning optimization algorithm. H Huang et al. [14] systematically analyzed the disturbances introduced by the tether and manipulator movement, then used them as feedforward terms to achieve precise motion control of observation-class ROV. Most of the controller mentioned above relies heavily on the dynamic model or disturbance observer. The control performance may degrade when the dynamic model parameters are inaccurate, or the manipulator disturbance cannot estimate accurately.

In contrast with the aforementioned methods, sliding mode control (SMC) is well-known for strong robustness to match uncertainties and perturbations. Thus, SMC is particularly well suited for the station-keeping control of ARV subject to manipulator disturbances, parameter uncertainties, and unknown external disturbances. Dannigan et al. [15] used a sliding mode approach to handle coupling effects induced by the manipulator. An improved nonsingular terminal SMC (I-NTSMC) scheme was designed for the coordinated control of the Underwater Biomimetic Vehicle-Manipulator System [16]. I-NTSMC scheme can provide strong robustness for the controller and alleviate the chattering effect in the meantime. Although the control scheme still uses manipulator dynamics information, its function is just to enhance the control performance, and the controller does not rely on it. Besides, there are also many other controllers based on SMC that have been successfully implemented in underwater vehicles in recent years [17–19].

The major drawback of the SMC is the chattering issue, which degrades the control performance, reduces the life of the actuator, and may excite high-frequency unmodeled dynamics. For these reasons, substantial methods on chattering reduction have been developed in recent years, such as the boundary layer method [20–22], adaptive control [23,24], fuzzy logic control [25,26], high-order SMC (HOSMC) [27–29]. Among these methods, the HOSMC method is the most effective chattering suppression method by hiding discontinuous terms behind an integrator [30]. Among the HOSM control method, the super-twisting algorithm (STA) is more popular due to being independent of higher-order sliding manifolds. As is described in [31], the STA is effective in suppressing the chattering while at the same time preserving the good properties of the first-order SMC. To improve the robustness and convergence velocity of STA when the system state is far away from the origin, Moreno [32] proposed the generalized super-twisting algorithm (GSTA), which includes extra linear correction terms compared to the standard STA. Besides, GSTA can also handle state-dependent disturbances, not just time-dependent disturbances [33]. However, the GSTA, as well as the STA, requires the upper bound information of the disturbance derivative, which is usually hard to obtain or overestimated in practice. Moreover, even the disturbances derivatives are time-varying, it will be desirable to follow their variation. To circumvent this problem, the variable gain strategy has been widely used. Recently, J. Guerrero et al. [34] developed an adaptive generalized super-twisting algorithm (AGSTA) for the trajectory tracking control of underwater vehicles, which does not require any prior knowledge about the upper bound of the disturbance derivative. However, the adaptive law used in this paper may still overestimate the control gains. Moreover, the controller required vehicle dynamic knowledge, which can not be attained accurately in practical applications.

In this paper, based on the previous results of J. Guerrero et al. [34], we proposed a modified AGSTA for the station-keeping control of the ARV. The ATD and RESO are used to enhance the controller performance. The main contributions of this paper are summarized as follows: (1) A new adaptive law is designed to adjust the gains of GSTA online, which effectively avoids overestimating control gains, and the control gains will not vibrate around the set minimum value. (2) The proposed controller is model-free and can be easily implemented in practical scenarios.

The remainder of this paper is organized as follows. Section 2 introduces ARV modeling, including vehicle dynamics and manipulator dynamics. In Section 3, the detailed design procedure of the proposed control scheme is presented, and the stability of the closed-loop system is analyzed based on Lyapunov's direct method. In Section 4, numeri-

cal simulation is performed to confirm the effectiveness of the proposed control scheme. Finally, some conclusions and potential future works are given in Section 5.

## 2. Model Dynamics

### 2.1. Vehicle Dynamics

The ARV dynamic model involves two reference frames, as shown in Figure 1. One is the inertial-fixed frame $\{n\}$, another is the body-fixed frame $\{b\}$ fixed at the center of buoyancy (COB) of the ARV. Adopting the SNAME notation [35], the vehicle's dynamic model considering the manipulator disturbances, hydrodynamic effects, unknown external disturbances can be written in a compact matrix form as [34]:

$$
\begin{aligned}
M\dot{v} + C(v)v + D(v)v + g(\eta) &= \tau_v + \tau_m + \tau_u \\
\dot{\eta} &= J(\eta)v
\end{aligned}
\tag{1}
$$

where $\eta = [x\, y\, z\, \phi\, \theta\, \psi]^T$ denotes the position and attitude vector in the inertial-fixed frame and $v = [u\, v\, w\, p\, q\, r]^T$ is the linear and angular velocities expressed in the body-fixed frame. $J(\eta) \in \mathbb{R}^{6\times6}$ is the Euler angle mapping matrix from the inertial-fixed frame to the body-fixed frame. $M \in \mathbb{R}^{6\times6}$ is the inertia matrix, including the effects of added mass, $C(v) \in \mathbb{R}^{6\times6}$ is the Coriolis and centripetal matrix, $D(v) \in \mathbb{R}^{6\times6}$ represents the hydrodynamic damping matrix, $g(\eta) \in \mathbb{R}^6$ is the vector of gravitational/buoyancy forces and moments expressed in the body-fixed frame. Finally, $\tau_m \in \mathbb{R}^6$ defines the vector of coupling reaction forces between the manipulator and the vehicle. $\tau_u \in \mathbb{R}^6$ represents the vector of unknown external disturbances. $\tau_v \in \mathbb{R}^6$ is the resultant vector of force/moment generated by thrusters. The vector of the thruster forces $u_T \in \mathbb{R}^6$ produced by each thruster can be calculated by:

$$
u_T = B^{-1}\tau_v
\tag{2}
$$

where $B \in \mathbb{R}^{6\times6}$ is a nonsingular thruster distribution matrix. Refer to Appendix B for further details about the thruster arrangement.

To simplify the vehicle dynamics model and reduce the computational burden, proper simplification should be made. As the Haidou-1 ARV usually works in the deep-sea circumstance, the effect of wave effect and the ocean current is not considered here. The interaction force between the optical fiber and the vehicle is negligible since it is very small and hard to model accurately. The hydrodynamic coefficients of Haidou-1 ARV are obtained by the computational fluid dynamics (CFD) method. Refer to Appendix A for further details about the vehicle dynamics model.

As the controller is normally designed in task space, it is necessary to establish the vehicle dynamics in the inertial-fixed frame. Using the following kinematic transformation equations (assuming that $J(\eta)$ is nonsingular):

$$
\begin{aligned}
M_\eta(\eta) &= J^{-T}(\eta)MJ^{-1}(\eta) \\
C_\eta(v,\eta) &= J^{-T}(\eta)\left[C(v) - MJ^{-1}(\eta)\dot{J}(\eta)\right]J^{-1}(\eta) \\
D_\eta(v,\eta) &= J^{-T}(\eta)D(v)J^{-1}(\eta) \\
\tau_{m\eta} &= J^{-T}(\eta)\tau_m \\
\tau_{u\eta} &= J^{-T}(\eta)\tau_u \\
g_\eta(\eta) &= J^{-T}(\eta)g(\eta) \\
\tau_\eta &= J^{-T}(\eta)\tau_v
\end{aligned}
\tag{3}
$$

The vehicle dynamics (1) can be rewritten in the inertial-fixed frame as:

$$
M_\eta(\eta)\ddot{\eta} + C_\eta(v,\eta)\dot{\eta} + D_\eta(v,\eta)\dot{\eta} + g_\eta(\eta) = \tau_\eta + \tau_{m\eta} + \tau_{u\eta}
\tag{4}
$$

By introducing a constant diagonal matrix $\hat{M}$, the vehicle dynamics (4) can be rewritten as:

$$
\ddot{\eta} = \hat{M}^{-1}\left(\tau_\eta + f(\eta,v)\right)
\tag{5}
$$

where $f(\eta, v)$ represents the total disturbance, including the internal nonlinear dynamics and external disturbance, and it is defined as follows:

$$f(\eta, v) = \tau_{m\eta} + \tau_{u\eta} - C_\eta(v, \eta)\dot{\eta} - D_\eta(v, \eta)\dot{\eta} - g_\eta(\eta) + (\hat{M} - M_\eta(\eta))\ddot{\eta} \tag{6}$$

To facilitate controller design, the following assumptions are required.

**Assumption 1.** *The time derivative of total disturbance $f(\eta, v)$ exists and is bounded.*

**Assumption 2.** *The pitch angle is restricted to $|\theta| < \pi/2$.*

Since the ARV is a mechanical system, $f(\eta, \nu)$ will not change infinitely fast. Therefore, assuming that $\dot{f}(\eta, \nu)$ is bounded is a valid assumption. According to Assumption 2, the inverse of the Jacobian matrix $J(\eta)$ exists. For the station-keeping operations, the desired pitch angle is 0 rad. The fully actuated vehicle has the ability to keep the pitch angle sufficiently far from $\pm\pi/2$ radian; besides, the vehicle has metacentric restoring forces. Thus, the pitch angle is unlikely to violate this restriction [36]. Assumption 2 is reasonable.

### 2.2. Manipulator Dynamics

In this paper, the Newton-Euler method was used to derive the dynamic model of the deep-sea manipulator. This method starts with a forward recursion algorithm that calculates the velocity and acceleration information of each link starting from the base and moving towards the end-effector [37]:

$$
\begin{aligned}
\omega^{i+1}_{i+1} &= R^{i+1}_i \omega^i_i + \dot{q}_{i+1}\hat{Z}_{i+1} \\
\dot{\omega}^{i+1}_{i+1} &= R^{i+1}_i \dot{\omega}^i_i + R^{i+1}_i \left( {}^i\omega_i \times \dot{q}_{i+1}{}^{i+1}\hat{Z}_{i+1} \right) + \ddot{q}_{i+1}\hat{Z}_{i+1} \\
v^{i+1}_{i+1} &= R^{i+1}_i \left( v^i_i + \omega^i_i \times P^i_{i+1} \right) \\
\dot{v}^{i+1}_{i+1} &= R^{i+1}_i \left( \dot{v}^i_i + {}^i\dot{\omega}_i \times P^i_{i+1} + \omega^i_i \times (\omega^i_i \times P^i_{i+1}) \right) \\
\dot{v}^{i+1}_{c,i+1} &= \dot{v}^{i+1}_{i+1} + \dot{\omega}^{i+1}_{i+1} \times P^{i+1}_{c,i+1} + \omega^{i+1}_{i+1} \times (\omega^{i+1}_{i+1} \times P^{i+1}_{c,i+1})
\end{aligned} \tag{7}
$$

where $q_{i+1}$ and $\dot{q}_{i+1}$ are the joint position and velocity of joint $i$, $z_{i+1} = [0\ 0\ 1]^T$, ${}^{i+1}R_i$ is the rotation matrix from frame $\{i\}$ to frame $\{i+1\}$, ${}^i\omega_i$ and ${}^i\dot{\omega}_i$ are the angular velocity and angular acceleration of link $i$ w.r.t. frame $\{i\}$. ${}^iv_i$ and ${}^i\dot{v}_i$ are the linear velocity and linear acceleration of link $i$ w.r.t. frame $\{i\}$. ${}^{i+1}\dot{v}_{c,i+1}$ represents the linear acceleration of the center of mass (COM) of link $i + 1$ expressed in frame $\{i + 1\}$. ${}^iP_{i+1}$ is the position vector from joint $i$ to joint $i + 1$ expressed in frame $\{i\}$. ${}^{i+1}P_{c,i+1}$ is the position vector from joint $i + 1$ to the COM of link $i + 1$ expressed in frame $\{i + 1\}$.

Then backward recursion algorithm performs the force and moment balance on each link and iterates backward from the end effector to the manipulator base. The generalized force and moment of link $i$ can be determined by:

$$
\begin{aligned}
{}^iF_i &= M_i \dot{v}_{c,i} \\
{}^iN_i &= I_i{}^i\dot{\omega}_i + {}^i\omega_i \times (I_i{}^i\omega_i)
\end{aligned} \tag{8}
$$

where $M_i$ and $I_i$ are the mass matrix and the inertia matrix of link $i$, respectively.

In the deep ocean environment, each link of the underwater manipulator will be affected by its self-motion and uniform flow. Due to unpredictable and weak ocean currents in the deep sea, only self-motion is taken into account here. The hydrodynamic forces exerted on the manipulator are similar to the vehicle, consisting of additional mass force, buoyancy, drag force, etc. As the manipulator moves slowly, the additional mass force acting on the link can be neglected.

The drag force acting on the link $i$ is calculated as [38]:

$$
\begin{aligned}
f_{hi} &= \tfrac{1}{2}\rho C_d D \int_0^{l_i} \|v(x)\| v(x) dx \\
n_{hi} &= \tfrac{1}{2}\rho C_d D \int_0^{l_i} \|v^n(x)\| \left\{ [x, 0, 0]^T \times v^n(x) \right\} dx
\end{aligned} \tag{9}
$$

where

$$v(x) = {}^i v_i + {}^i \omega_i \times [x, 0, 0] = [v_1(x), v_2(x), v_3(x)]^T$$
$$v^n(x) = [0, v_2(x), v_3(x)]^T$$

where $\rho$ is the density of seawater, $C_d$ is the drag coefficient, $C_m$ is the inertia force coefficient, $A$ is the projected area that is perpendicular to the velocity of the incoming flow, $D$ is the cross-sectional area of the object, $l_i$ is the length of link $i$, the empirical values of $C_d = 1.0$ is used here [39].

The vectors of the interaction of forces and moments between two adjacent links are calculated as follows:

$$
\begin{aligned}
{}^i f_i &= {}^i R_{i+1}{}^{i+1} f_{i+1} + {}^i F_i + m_i g_i + b_i + f_{hi} \\
{}^i n_i &= {}^i R_{i+1}{}^{i+1} n_{i+1} + {}^i N_i + {}^i P_{i+1} \times ({}^i R_{i+1}{}^{i+1} f_{i+1}) + P_{ci}^i \times {}^i F_i - P_{ci}^i \times m_i g - P_{bi}^i \times b_i + n_{hi}
\end{aligned}
\tag{10}
$$

where $P_{bi}^i$ is the position vector from the COB of link $i$ to the origin of frame $\{i\}$. $m_i$ is the mass of the link $i$, $b_i$ is the vector of buoyancy of link $i$.

The dynamic coupling force/moment acting on the vehicle can be calculated as follows:

$$
\tau_m = \begin{bmatrix} {}^b R_0{}^0 f_0 \\ {}^b R_0{}^0 n_0 + {}^b P_0 \times ({}^b R_0{}^0 f_0) \end{bmatrix}
\tag{11}
$$

where $f_0^0$ and $t_0^0$ are the vectors of the force and the moment exerted on the manipulator base. $P_0^b$ is the position vector from the body-fixed frame $\{b\}$ to the manipulator base frame $\{0\}$.

Due to the particular operating environment and irregular geometric shape, the dynamics of the deep-sea manipulator are more complicated than on-shore industrial manipulators. The hydrodynamic coefficients are difficult to measure or estimate accurately, and the joint friction term is also hard to model accurately owing to the outside seawater pressure. Furthermore, the manipulator dynamics involve the iterative algorithm, which is computationally expensive. Thus, the control scheme should not rely on the dynamic model of the deep-sea manipulator.

## 3. Controller Design

In this section, the design of the proposed control scheme for the station-keeping control of ARVs is addressed. The control objective is to stabilize the vehicle in a desired position and attitude while the manipulator performs the manipulation task. Inspired by the methodology presented in [34], we proposed a new adaptive law for the GSTA. The ATD and RESO are used to enhance the control performance of modified AGSTA. Figure 2 presents the control block diagram.

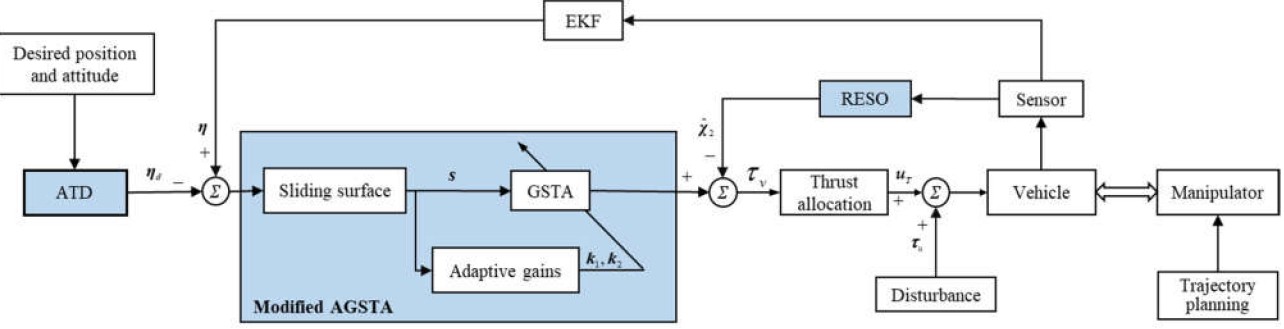

**Figure 2.** Diagram of the station-keeping control framework.

### 3.1. Adaptive Tracking Differentiator

In the free-floating manipulation, the vehicle has difficulty tracking the fast change reference signal due to larger inertia. Besides, the differential signal is usually extracted by the backward difference method, which is very sensitive to noise. To overcome these problems, Cai, Mingxue, et al. [16] proposed the ATD, which can provide the reference signal and its derivative with a smooth and adjustable process. Compared to the conventional tracking differentiator, ATD enhances the ability to adapt different step signals with the variable filter coefficient and fast coefficient.

The discrete form of the ATD can be designed as follows:

$$
\begin{cases}
r_0 = A/4T^2 \\
h_0 = \begin{cases} 4h & \text{without noise} \\ (A/A_n + 1)h & \text{with noise} \end{cases} \\
y_1(k+1) = y_1(k) + hy_2(k) \\
y_2(k+1) = y_2(k) + hfhan(y_1(k) - x(k), y_2(k), r_0, h_0)
\end{cases}
\tag{12}
$$

where $T$ is the desired transient time of the reference signal, $A$ is the amplitude of step signal, $h$ is the sampling period, $r_0$ is the fast coefficient, $h_0$ is the filter coefficient, $A_n$ denotes the amplitude of the noise signal, $x(k)$ is the input signal of ATD, $y_1(k)$ and $y_2(k)$ represent the target reference signal and corresponding derivative, respectively. $fhan(\cdot)$ is denoted as the optimal synthetic rapid control function which is proposed as:

$$
\begin{cases}
a_0 = r_0 h_0^2 \\
a_1 = h_0 x_2 \\
a_2 = x_1 + a_1 \\
a_3 = \sqrt{a_0(a_0 + 8|a_2|)} \\
b_0 = a_1 + sign(a_2)(a_3 - a_0)/2 \\
b_1 = [sign(a_2 + a_0) - sign(a_2 - a_0)]/2 \\
b_2 = (a_1 + a_2 - b_0)b_1 + b_0 \\
b_3 = [sign(b_2 + a_0) - sign(b_2 - a_0)]/2 \\
fhan(x_1, x_2, r_0, h_0) = -r_0[b_2/a_0 - sign(b_2)]b_3 - r_0 sign(b_2)
\end{cases}
\tag{13}
$$

### 3.2. Reduced-Order Extended State Observer

In this section, an extended state observer (ESO) is developed to estimate the total disturbance term in real-time. Considering that the system state variables of ARV can be measured relatively accurately, there is no need to estimate it. Besides, the initial estimation error will cause an initial peaking phenomenon. Therefore, a reduced-order ESO is preferred.

To use RESO, we introduce a change of variables. Define $\chi_1 = \dot{\eta}$, $\chi_2 = \hat{M}^{-1} f(\eta, v)$.

Note that Assumption 1 indicates that the time derivative of the total disturbance term $f(\eta, v)$ exists. Therefore, the vehicle dynamics (5) can be extended as the following state-space form:

$$
\begin{cases}
\dot{\chi}_1 = \hat{\chi}_2 + \hat{M}^{-1} \tau_\eta \\
\dot{\chi}_2 = \hat{M}^{-1} \dot{f}(\eta, v)
\end{cases}
\tag{14}
$$

Defined $\hat{\chi}_1$ and $\hat{\chi}_2$ as the estimated value of the state variables. The corresponding estimation error is defined as $\tilde{\chi}_i = \chi_i - \hat{\chi}_i$, $i = 1, 2$. Then the RESO is designed as [40]:

$$
\begin{cases}
\dot{\hat{\chi}}_1 = \hat{\chi}_2 + \hat{M}^{-1} \tau_\eta + \beta_1 \tilde{\chi}_1 \\
\dot{\hat{\chi}}_2 = \beta_2 \tilde{\chi}_1
\end{cases}
\tag{15}
$$

where $\beta_i \in \mathbb{R}^{6 \times 6}$, $i = 1$, 2, are the corresponding observer gains. Using the bandwidth-parameterization method proposed in [41], all observer poles can be placed at $-\omega_i$. To this end, the observer gains can be selected as follows:

$$\begin{cases} \beta_1 = diag(2\omega_1, 2\omega_2, 2\omega_3, 2\omega_4, 2\omega_5, 2\omega_6) \\ \beta_2 = diag(\omega_1, \omega_2, \omega_3, \omega_4, \omega_5, \omega_6) \end{cases} \tag{16}$$

As a result, the RESO has only one parameter to be tuned in one DOF, which is the observer bandwidth. It should be noted that a smaller estimation error can be obtained by increasing the bandwidth of RESO. However, the RESO will become more sensitive to the measurement noise. Therefore, the observer bandwidth should be selected to achieve a trade-off between the observer performance and the capability of noise tolerance.

*3.3. Modified AGSTA Design*

There exist two phases for the SMC, i.e., the sliding phase and reaching phase. For the sliding phase, a linear sliding surface is chosen as follows:

$$s = \dot{e} + ce \tag{17}$$

where $s = [s_1, \ s_2, \ \dots \ s_6]^T$, $e = \eta - \eta_d$ is the error vector. $c = diag(c_1, c_2 \cdots c_6)$ is a diagonal positive definite matrix, which determines the convergence rate of the error.

To obtain good control performance during the reaching phase and alleviate chattering, the following GSTA will be used:

$$\begin{aligned} \dot{s}_i &= -k_{1i}\phi_{1i}(s_i) + \varpi_i \\ \dot{\varpi}_i &= -k_{2i}\phi_{2i}(s_i) \end{aligned} \tag{18}$$

where

$$\phi_{1i}(s_i) = \mu_{1i}|s_i|^{\frac{1}{2}}sgn(s_i) + \mu_{2i}s_i$$
$$\phi_{2i}(s_i) = \tfrac{1}{2}\mu_{1i}^2 sgn(s_i) + \tfrac{3}{2}\mu_{1i}\mu_{2i}|s_i|^{\frac{1}{2}}sgn(s_i) + \mu_{2i}^2 s_i$$

One can see that for the choice $\mu_{2i} = 0$, the structure of the standard STA is recovered. The linear growth term $\mu_{2i}s_i$ in $\phi_{1i}(s_i)$ helps to counteract the effects of state-dependent perturbations, which can exponentially increase in time.

The adaptive gains $k_{1i}(t)$ and $k_{2i}(t)$ are updated as:

$$\begin{aligned} \dot{k}_{1i}(t) &= \begin{cases} \ell_i & |s_i| > \sigma_i \\ -r_i & k_{1i} > k_{1i\min}, |s_i| \le \sigma_i \\ 0 & k_{1i} \le k_{1i\min}, |s_i| \le \sigma_i \end{cases} \\ k_{2i}(t) &= 2\varepsilon_i k_{1i} + \beta_i + 4\varepsilon_i^2 \end{aligned} \tag{19}$$

where $\mu_{1i}$, $\mu_{2i}$, $\omega_i$, $r_i$, $\sigma_i$, $\beta_i$, $k_{1i\min}$, $\sigma_i$ and $\varepsilon_i$ are arbitrary positive constants. $r_i$ is utilized to adjust the decrease rate of $k_{1i}$, which avoids overestimating the control gains. Note that $r_i$ is set to $\ell_i$ in many papers [42–44], which may not be the best choice. $\sigma_i$ denotes the predefined neighborhood of zero, smaller $\sigma_i$ can ensure lower steady-state error but increase the chattering effect as well. $k_{1i\min}$ is used to maintain the control performance when the sliding variable converges to the region $|s_i| < \sigma_i$. When $k_{1i} \le k_{1i\min}$ and $|s_i| \le \sigma_i$, $\dot{k}_{1i}(t)$ is set to 0 instead of $\ell_i$, which avoids the control gains are varying in a zigzag motion.

Finally, combining the RESO and modified AGSTA, the control scheme for the station-keeping control of ARV is given as follows:

$$\tau_v = J^T \hat{M}[\ddot{\hat{\eta}}_d + c\dot{e} - \hat{\chi}_2 + v] \tag{20}$$

where $\hat{\ddot{\eta}}_d$ denotes the nominal value of the second derivative of the reference signal, $v = [v_1, v_2, \cdots, v_6]^T$, and each element of this vector is given as:

$$v_i = -k_{1i}\phi_{1i}(s_i) - \int k_{2i}\phi_{2i}(s_i) \tag{21}$$

It should be noted that we cannot obtain $\ddot{\eta}_d$ with the ATD. Nevertheless, we can replace it with a nominal value, such as 0. The corresponding error will be treated as an unknown disturbance.

*3.4. Stability Analysis*

The following lemma is used for the stability analysis.

**Lemma 1.** [45]. *Consider a nonlinear system $\dot{x} = f(t, x)$. Suppose there exists a continuous positive definite function $V(x)$, which satisfies $\dot{V}(x) \leq -\lambda V^\alpha(x)$. Then, the system state arrives at the equilibrium point in a finite-time $t_s$ which satisfies $t_s \leq V^{(1-\alpha)}(0)/\lambda(1-\alpha)$.*

Substituting the controller (20) into the vehicle dynamics (5) yields the closed-loop dynamics, then take the *i*-th DOF to analyze for convenience; we have

$$\begin{aligned} \dot{s} &= -k_1[\mu_1|s|^{\frac{1}{2}}sgn(s) + \mu_2 s] + \varpi \\ \dot{\varpi} &= -k_2[\frac{1}{2}\mu_1^2 sgn(s) + \frac{3}{2}\mu_1\mu_2|s|^{\frac{1}{2}}sgn(s)] + \dot{h}(t) \end{aligned} \tag{22}$$

where $h_i(t) = [\tilde{\chi}_2 + \ddot{\eta}_d - \hat{\ddot{\eta}}_d]_i$ stands for the estimation error of RESO and $\ddot{\eta}_d$ which has been proved to be bounded in [40].

Without loss of generality, Equation (22) can be rewritten in a compact form:

$$\begin{aligned} \dot{s} &= -k_1\phi_1(s) + \varpi \\ \dot{\varpi} &= -k_2\phi_2(s) + \dot{h} \end{aligned} \tag{23}$$

where

$$\begin{aligned} \phi_1(s) &= \mu_1|s|^{\frac{1}{2}}sgn(s) + \mu_2 s \\ \phi_2(s) &= \frac{1}{2}\mu_1^2 sgn(s) + \frac{3}{2}\mu_1\mu_2|s|^{\frac{1}{2}}sgn(s) \end{aligned}$$

For the convenience of Lyapunov analysis, a new state vector is introduced here:

$$z = [z_1\ z_2]^T = [\phi_1(s)\ \varpi]^T \tag{24}$$

Noting that $\phi_2(s) = \phi_1'(s)\phi_1(s)$ and $\phi_1'(s) = \mu_1/2|s|_{\frac{1}{2}} + \mu_2$. Then, the time derivative of $z$ is obtained as:

$$\begin{aligned} \dot{z}_1 &= \phi_1'(s)[-k_1 z_1 + z_2] \\ \dot{z}_2 &= -\phi_1'(s)k_2 z_1 + \dot{h} \end{aligned} \tag{25}$$

where $\dot{h}$ is bounded as $\left|\dot{h}\right| \leq h_b$. Thus, there exists an unknown bounded function $L(t)$ satisfying $\dot{h} = L(t)\phi_1'(s)$.

**Remark 1.** *Note that the differential equations in (25) have discontinuous right-hand sides; the solutions of this differential inclusion are understood in Filippov's sense [46].*

Equation (25) can be rewritten in a vector-matrix format:

$$\dot{z} = \phi_1'(s)Az \tag{26}$$

where

$$A = \begin{bmatrix} -k_1 & 1 \\ -k_2 + L(t) & 0 \end{bmatrix}$$

Then, the following Lyapunov candidate function will be used to prove the closed-loop stability.

$$V(z_1, z_2, k_1, k_2) = V_0(z) + \frac{1}{2\varsigma_1}(k_1 - k_1^*)^2 + \frac{1}{2\varsigma_2}(k_2 - k_2^*)^2 \tag{27}$$

where $\varsigma_1$, $\varsigma_2$, $k_1^*$ and $k_2^*$ are positive constants and $V_0(z)$ is given by:

$$V_0(z) = (\beta + 4\varepsilon^2)z_1^2 - 4\varepsilon z_1 z_2 + z_2^2 = z^T P z \tag{28}$$

$$P = P^T = \begin{bmatrix} \beta + 4\varepsilon^2 & -2\varepsilon \\ -2\varepsilon & 1 \end{bmatrix} > 0$$

Furthermore, it is easy to verify that $V(\cdot)$ is positive definite and radially unbounded. It is noteworthy that $P$ is a positive definite matrix if $\beta$ is an arbitrary positive constant. By using the standard inequality for quadratic forms, we obtain:

$$\lambda_{\min}\{P\}\left|\left|z\right|\right|_2^2 \le V_0(z) \le \lambda_{\max}\{P\}\left|\left|z\right|\right|_2^2 \tag{29}$$

where $\left|\left|z\right|\right|_2^2 = \mu_1^2\left|s\right| + 2\mu_1\mu_2\left|s\right|_{\frac{3}{2}} + \mu_2^2 s^2 + \omega^2$ represents the Euclidean norm. $\lambda_{\min}\{\cdot\}$ and $\lambda_{\max}\{\cdot\}$ are the minimum and maximum eigenvalues of matrices, respectively. Further, we can derive the following inequality:

$$|s|^{\frac{1}{2}} \le \frac{1}{\mu_1}||z||_2 \le \frac{V_0^{\frac{1}{2}}(z)}{\mu_1 \lambda_{\min}^{\frac{1}{2}}\{P\}} \tag{30}$$

The time derivative of $V_0(z)$ is further given as:

$$\dot{V}_0(z) = \dot{z}^T P z + z^T P \dot{z} = \phi_1'(s)z^T[A^T P + PA]z = -\phi_1'(s)z^T Q z \tag{31}$$

where

$$Q = \begin{bmatrix} 2k_1(\beta + 4\varepsilon^2) - 4\varepsilon(k_2 - L(t)) & k_2 - \beta - 4\varepsilon^2 - 2\varepsilon k_1 - L(t) \\ k_2 - \beta - 4\varepsilon^2 - 2\varepsilon k_1 - L(t) & 4\varepsilon \end{bmatrix}$$

Selecting the gain $k_2 = \beta + 2\varepsilon k_1 + 4\varepsilon^2$, the matrix $\boldsymbol{Q}$ will be a positive definite matrix with minimum eigenvalue $2\varepsilon$ if $k_1$ satisfies the following inequality:

$$k_1 > \frac{L^2(t)}{4\varepsilon\beta} + \frac{2\varepsilon(\beta + 4\varepsilon^2 - L) + \varepsilon}{\beta} \tag{32}$$

Then, the time derivative of $V_0(z)$ satisfies the following inequalities

$$\dot{V}_0(z) = -\phi_1'(s)z^T Q z \le -2\varepsilon(\mu_1/2\left|s\right|_{\frac{1}{2}} + \mu_2)\left|\left|z\right|\right|_2^2 \tag{33}$$

Substituting (29) and (30) into (33), we can obtain:

$$\dot{V}_0(z_i) \le -\frac{\mu_1^2\varepsilon}{\lambda_{\min}^{\frac{1}{2}}\{P\}}V_0^{\frac{1}{2}}(z) - 2\mu_2\varepsilon\frac{V_0(z)}{\lambda_{\max}\{P\}} \le -\wp V_0^{\frac{1}{2}}(z) \tag{34}$$

where

$$\wp = \frac{\mu_1^2\varepsilon}{\lambda_{\min}^{\frac{1}{2}}\{P\}}$$

Differentiating the Lyapunov function candidate (27) w.r.t. time yields:

$$
\begin{aligned}
\dot{V}(\cdot) &= \dot{V}_0(z) + \tfrac{1}{\varsigma_1}\left(k_1 - k_1^*\right)\dot{k}_1 + \tfrac{1}{\varsigma_2}\left(k_2 - k_2^*\right)\dot{k}_2 \\
&\le -\wp V_0^{\frac{1}{2}}(z) + \tfrac{1}{\varsigma_1}\left(k_1 - k_1^*\right)\dot{k}_1 + \tfrac{1}{\varsigma_2}\left(k_2 - k_2^*\right)\dot{k}_2 \\
&= -\wp V_0^{\frac{1}{2}}(z) - \tfrac{\omega_1}{\sqrt{2\varsigma_1}}\left|k_1 - k_1^*\right| - \tfrac{\omega_2}{\sqrt{2\varsigma_2}}\left|k_2 - k_2^*\right| + \tfrac{1}{\varsigma_1}\left(k_1 - k_1^*\right)\dot{k}_1 \\
&\quad + \tfrac{1}{\varsigma_2}\left(k_2 - k_2^*\right)\dot{k}_2 + \tfrac{\omega_1}{\sqrt{2\varsigma_1}}\left|k_1 - k_1^*\right| + \tfrac{\omega_2}{\sqrt{2\varsigma_2}}\left|k_2 - k_2^*\right|
\end{aligned}
\tag{35}
$$

By using the well-known inequality $\sqrt{x^2 + y^2 + z^2} \le |x| + |y| + |z|$ on (35), the following inequality is obtained

$$
-\wp V_0^{\frac{1}{2}}(z_i) - \frac{\omega_1}{\sqrt{2\varsigma_1}}\left|k_1 - k_1^*\right| - \frac{\omega_2}{\sqrt{2\varsigma_2}}\left|k_2 - k_2^*\right| \le -\hbar\sqrt{V(\cdot)}
\tag{36}
$$

where

$$
\hbar = \min(\wp, \omega_1, \omega_2)
$$

As indicated in [47], $k_1$ and $k_2$ are bounded. Therefore, there exist positive constants $k_1^*$ and $k_2^*$ satisfying $k_1 - k_1^* < 0$ and $k_2 - k_2^* < 0$ for $\forall t \ge 0$.

Then, the time derivative of $V(z_1, z_2, k_1, k_2)$ can be rewritten as:

$$
\begin{aligned}
\dot{V}(\cdot) &\le -\hbar\sqrt{V(\cdot)} - \left|k_1 - k_1^*\right|\left(\tfrac{1}{\varsigma_1}\dot{k}_1 - \tfrac{\omega_1}{\sqrt{2\varsigma_1}}\right) - \left|k_2 - k_2^*\right|\left(\tfrac{1}{\varsigma_2}\dot{k}_2 - \tfrac{\omega_2}{\sqrt{2\varsigma_2}}\right) \\
&= -\hbar\sqrt{V(\cdot)} + \vartheta
\end{aligned}
\tag{37}
$$

where

$$
\vartheta = -\left|k_1 - k_1^*\right|\left(\frac{1}{\varsigma_1}\dot{k}_1 - \frac{\omega_1}{\sqrt{2\varsigma_1}}\right) - \left|k_2 - k_2^*\right|\left(\frac{1}{\varsigma_2}\dot{k}_2 - \frac{\omega_2}{\sqrt{2\varsigma_2}}\right)
$$

Finally, some specific conditions should be discussed for accomplishing the proof. For further analysis of $\dot{V}(\cdot)$, its sign should be discussed in three conditions.

When $|s| > \sigma$, $\dot{k}_1 = \ell$, and the term $\vartheta$ is computed as:

$$
\vartheta = -\left|k_1 - k_1^*\right|\left(\frac{1}{\varsigma_1}\ell - \frac{\omega_1}{\sqrt{2\varsigma_1}}\right) - \left|k_2 - k_2^*\right|\left(\frac{1}{\varsigma_2}2\varepsilon\ell - \frac{\omega_2}{\sqrt{2\varsigma_2}}\right)
\tag{38}
$$

By selecting $\ell = \omega_1\sqrt{\varsigma_1/2}$ and $\varepsilon = \omega_2\sqrt{\varsigma_2/2}/2\ell$, the term $\vartheta = 0$ can be met. Then, Equation (37) can be rewritten as:

$$
\dot{V}(\cdot) \le -\hbar\sqrt{V(\cdot)}
\tag{39}
$$

The gain $k_1$ will increase monotonically based on adaptive law until the condition (32) is reached. Then the sliding variable will reach the domain $|s| \le \sigma$ in the finite time according to Lemma 1.

When $k_{1i} > k_{1i\min}$ and $|s_i| \le \sigma_i$, $\dot{k}_1 = -r$, $\vartheta$ is computed as:

$$
\vartheta = \left|k_1 - k_1^*\right|\left(\frac{1}{\varsigma_1}r + \frac{\omega_1}{\sqrt{2\varsigma_1}}\right) + \left|k_2 - k_2^*\right|\left(\frac{1}{\varsigma_2}2\varepsilon r + \frac{\omega_2}{\sqrt{2\varsigma_2}}\right)
\tag{40}
$$

It is obvious that $\dot{k}_1 < 0$ and $\vartheta > 0$. Thus the sign of $\dot{V}(\cdot)$ is uncertain, which means $|s|$ may become greater than $\sigma$ with the decrease of $k_1$. Once the sliding variable exceeds this domain, it will return to the domain $|s| \le \sigma$ in finite time, as discussed above. It is noted that the condition of $k_1 \le k_{1\min}$ and $|s| \le \sigma$ is similar to that of $k_1 > k_{1\min}$ and $|s| \le \sigma$. Hence we omit the corresponding discussion.

Finally, we can conclude that the sliding variable will converge to the predefined neighborhood of origin in the finite time and remain in a larger domain $|s| \le \bar{\sigma}, \bar{\sigma} > \sigma$ for all subsequent time. Since we use a linear sliding surface, the error vector will confine to

the neighborhood of origin asymptotically. The whole system is stable and bounded in finite time.

## 4. Simulation Results

To validate the feasibility and efficiency of the proposed control scheme, simulations will be conducted on a realistic model Haidou-1 ARV in the MATLAB/Simscape software environment.

### 4.1. Description of the Simulation System

The ARV model considered in the simulation is composed of a 6 DoFs vehicle and a 6 DoFs deep-sea electric manipulator and is operating in the 11,000 m deep sea. The vehicle has six thrusters in total to provide the station-keeping capabilities, including two vertical thrusters, two side thrusters, and two rotatable thrusters (in the stern side). The thrust force saturation is considered in simulation, and the thrust force can range from −146.9 N to 231.6 N. The vehicle can be fully actuated when the manipulator performs underwater manipulation tasks.

The manipulator mounted on the bow part of the vehicle is a 7-function (with 6 DoFs and 1 clamping function) electric manipulator that can operate in 11,000 m deep sea. The Denavit-Hartenberg (D-H) parameters of the deep-sea electric manipulator are specified in Table 1, and the manipulator weighs 36.6 kg in water.

**Table 1.** D-H parameters for the deep-sea electric manipulator.

| Joint $i$ | $a_{i-1}$ (mm) | $d_i$ (mm) | $\alpha_{i-1}$ (deg) | Joint Limit (deg) |
|---|---|---|---|---|
| 1 | 0 | 0 | 90 | −30~90 |
| 2 | 130.4 | 138.22 | 90 | −30~90 |
| 3 | 641.8 | 0 | 0 | −90~30 |
| 4 | 220.2 | 442.64 | 90 | −90~90 |
| 5 | 21.5 | 0 | −90 | 0~90 |
| 6 | −48.1 | 239.60 | 90 | −180~180 |

To make the simulation more consistent with the realistic scenario. The sensor measurement noises should be taken into consideration in the simulation. The following sensor noises are added: zero-mean Gaussian noises with the variance of $1 \times 10^{-4}$ are added to the ideal vehicle position signals; zero-mean Gaussian noises with the variance of $2.74 \times 10^{-5}$ are added to the ideal vehicle attitude signals; zero-mean Gaussian noises with the variance of $2.5 \times 10^{-7}$ are added to the ideal vehicle velocity signals. The measurement signals are handled by an extended Kalman filter (EKF) and then used as the controller's input signals, as shown in Figure 2.

Moreover, the parameters of the dynamic model used in the controller are assumed to be inaccurate and have about 30% parameter uncertainties. In this study, the unknown time-varying external disturbances are simulated using the periodic functions as follows:

$$
d_u = \begin{cases}
5 + 3\sin(0.2t + \pi/5)N \\
5 + 4\cos(0.3t + \pi/6)N \\
10 + 8\cos(0.2t)N \\
25 + 16\sin(0.2t)N \cdot m \\
30 + 15\sin(0.2t + \pi/4)N \cdot m \\
8 + 6\cos(0.3t)N \cdot m
\end{cases}
\tag{41}
$$

Finally, the control scheme is discretized by the explicit Euler method, and the sampling period is set to 0.1 s.

### 4.2. Description of the Task

In the simulation task, the vehicle performs station-keeping while the manipulator is used to grasp the stationary target. The desired grasping process is illustrated in Figure 3. This simulation is used to verify the station-keeping ability of the ARV after using the proposed controller. In the simulation, the vehicle's initial state is set as: $\eta_0 = [0.02, 0.02, 0.03, \pi/90, \pi/90, \pi/90]^T$ and $\nu_0 = [0.002, 0.001, 0.002, 0.002, 0.002, 0.001]^T$. The manipulator's initial state is set to $q_0 = [90°, 90°, -90°, 0°, 90°, 0°]^T$. The desired value $\eta_d$ and $\nu_d$ are all set to 0. The fifth-order polynomial curves are used to plan joint trajectories in the joint space, and the desired joint trajectories are given in Figure 4. The contact process between the gripper and the object to be manipulated is not considered here. We only linearly added the weight of the grabbed target in water to the manipulator's end-effector in 0.5 s.

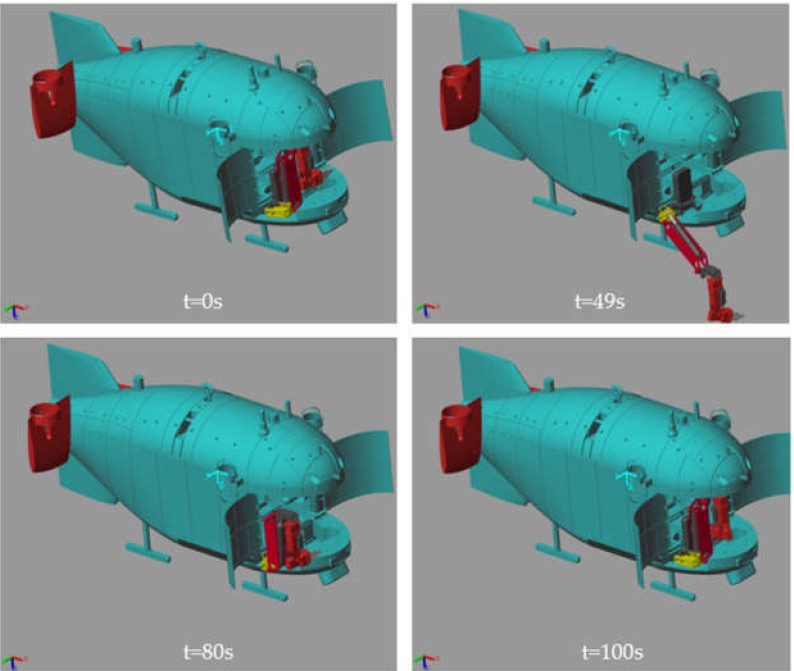

**Figure 3.** The process of free-floating manipulation during simulation.

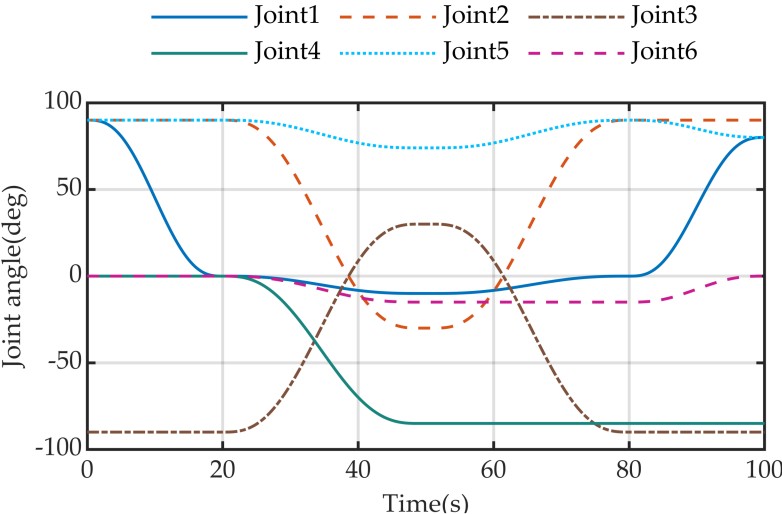

**Figure 4.** Planned joint trajectories.

Since our work is based on the previous results of J. Guerrero et al. [34] and can therefore be used as benchmarks of the performance of the proposed controller. To evaluate the control performance quantitatively, two performance indexes are defined as follows:

$$CHAT = \frac{1}{N} \sum_{k=0}^{N-1} \left| \tau_{v,k+1} - \tau_{v,k} \right| \tag{42}$$

$$RMSE = \sqrt{\frac{1}{N} \sum_{k=0}^{N-1} e_k^2} \tag{43}$$

where $t_{k+1} - t_k = h > 0$ denotes the sampling period, and we denote $f(t_k) = f_k$ with $k = 0, 1, 2 \dots N$. The first performance index calculates the average variation of the control input signals, which can evaluate the chattering effect quantitatively [48]. The second objective function represents the root mean squared error (RMSE), which can assess the control performance. These two performance indexes allow us to trade off chattering reduction ability and tracking control performance of the controller.

The parameter tuning of the controller is treated as a multi-objective optimization problem, and we use the gamultiobj function in MATLAB to solve it. The chosen control parameters are listed in Table 2.

**Table 2.** Controller parameters for the simulation.

| AGSTA | $[c_i, \mu_{1i}, \mu_{2i}, \ell_i, \varepsilon_i, \beta_i, k_{1i}(0), \sigma_i]$ |
|---|---|
| $x$ | [2.5500, 0.0021, 0.0690, 0.3192, 3.1620, 1.2692, 0.0049, 0.0020] |
| $y$ | [1.5830, 0.0023, 0.1189, 0.7272, 2.1222, 2.5352, 0.0040, 0.0028] |
| $z$ | [5.1820, 0.0031, 0.1628, 0.3680, 1.8180, 1.2795, 0.0063, 0.0066] |
| $\phi$ | [5.7932, 0.0032, 0.2474, 1.1557, 2.0171, 0.9480, 0.0066, 0.0155] |
| $\theta$ | [4.0165, 0.0030, 0.0609, 0.5230, 3.9955, 3.2493, 0.0080, 0.0079] |
| $\psi$ | [2.8460, 0.0020, 0.2053, 0.7632, 1.4879, 0.6179, 0.0065, 0.0055] |
| **Proposed** | $[c_i, \mu_{1i}, \mu_{2i}, \ell_i, r_i, k_{1i}(0), k_{1\text{mini}}, \varepsilon_i, \beta_i, \sigma_i, \omega_i]$ |
| $x$ | [2.0576, 0.0023, 0.1315, 1.4140, 0.8028, 0.0045, 0.4425, 1.1732, 2.1554, 0.0017, 1.0792] |
| $y$ | [1.0407, 0.0029, 0.2787, 1.1101, 0.9778, 0.0066, 0.6360, 0.6514, 0.7757, 0.0019, 1.1541] |
| $z$ | [3.1099, 0.0065, 0.2468, 1.4638, 0.2408, 0.0050, 0.5251, 0.6666, 1.5306, 0.0031, 0.5379] |
| $\phi$ | [8.1867, 0.0068, 0.1687, 0.7613, 0.5755, 0.0658, 0.3288, 1.9506, 2.5442, 0.0077, 1.0778] |
| $\theta$ | [3.1722, 0.0269, 0.1682, 0.6937, 0.6108, 0.0068, 0.1132, 0.2298, 1.4234, 0.0032, 1.1901] |
| $\psi$ | [2.3681, 0.0026, 0.1639, 0.8678, 0.5303, 0.0061, 0.4155, 1.6172, 2.6475, 0.0015, 0.5143] |

*4.3. Results and Discussions*

Figure 5 depicts the time responses of vehicle state variables during station-keeping control, and Figure 6 displays the corresponding errors. From Figure 5, one can notice that the desired reference signals generated by the ATD are smooth and are reached for both controllers. However, the origin AGSTA controller has a more significant overshoot and steady-state error than the proposed controller. The RMSE values are given in Table 3, which indicates that the proposed controller has smaller RMSE values than the origin AGSTA controller.

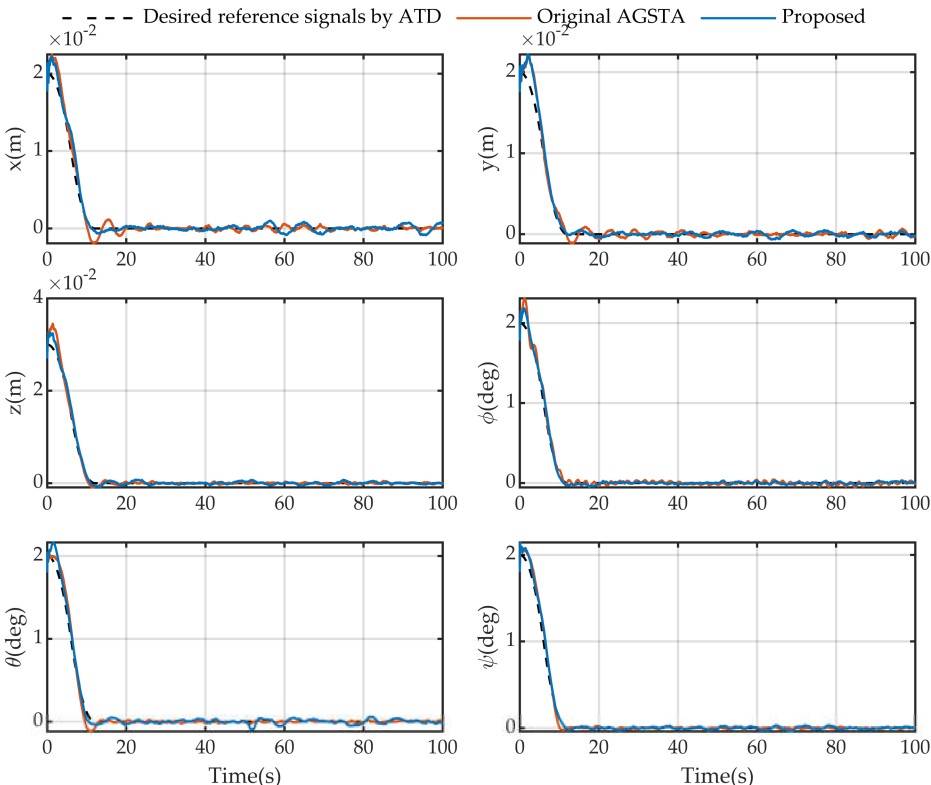

**Figure 5.** Time response of the vehicle state variables during station-keeping control.

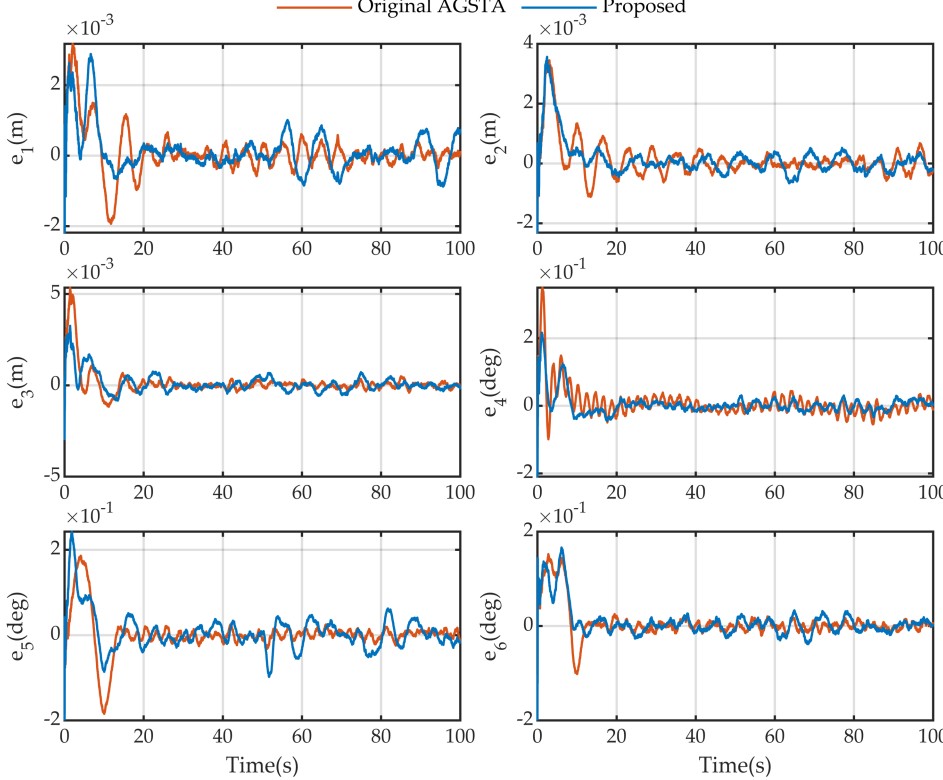

**Figure 6.** Vehicle state errors during station-keeping control.

**Table 3.** Comparison of performance indices.

| State Variables | CHAT | | RMSE | |
|:---:|:---:|:---:|:---:|:---:|
| | AGSTA | Proposed | AGSTA | Proposed |
| $x$ | 0.0363 | 0.0285 | $6.1005 \times 10^{-4}$ | $5.9895 \times 10^{-4}$ |
| $y$ | 0.0749 | 0.0462 | $6.0119 \times 10^{-4}$ | $5.8173 \times 10^{-4}$ |
| $z$ | 0.1045 | 0.0672 | $7.5175 \times 10^{-4}$ | $5.1327 \times 10^{-4}$ |
| $\phi$ | 0.0903 | 0.0538 | $7.5865 \times 10^{-4}$ | $5.3599 \times 10^{-4}$ |
| $\theta$ | 0.1173 | 0.0887 | $7.7607 \times 10^{-4}$ | $7.5049 \times 10^{-4}$ |
| $\psi$ | 0.1006 | 0.0662 | $5.6136 \times 10^{-4}$ | $5.5314 \times 10^{-4}$ |

Figure 7 shows the thrust force for each thruster during the simulation. As expected, the thrust forces of the original AGSTA controller exist serious chattering phenomenon. In contrast with the original AGSTA controller, the proposed controller produces a relatively smooth control input which proves the proposed controller has a better chattering reduction ability. A quantitative performance comparison of the chattering is also given in Table 3. The CHAT value of our proposed controller is 61.8%~78% of the original AGSTA controller.

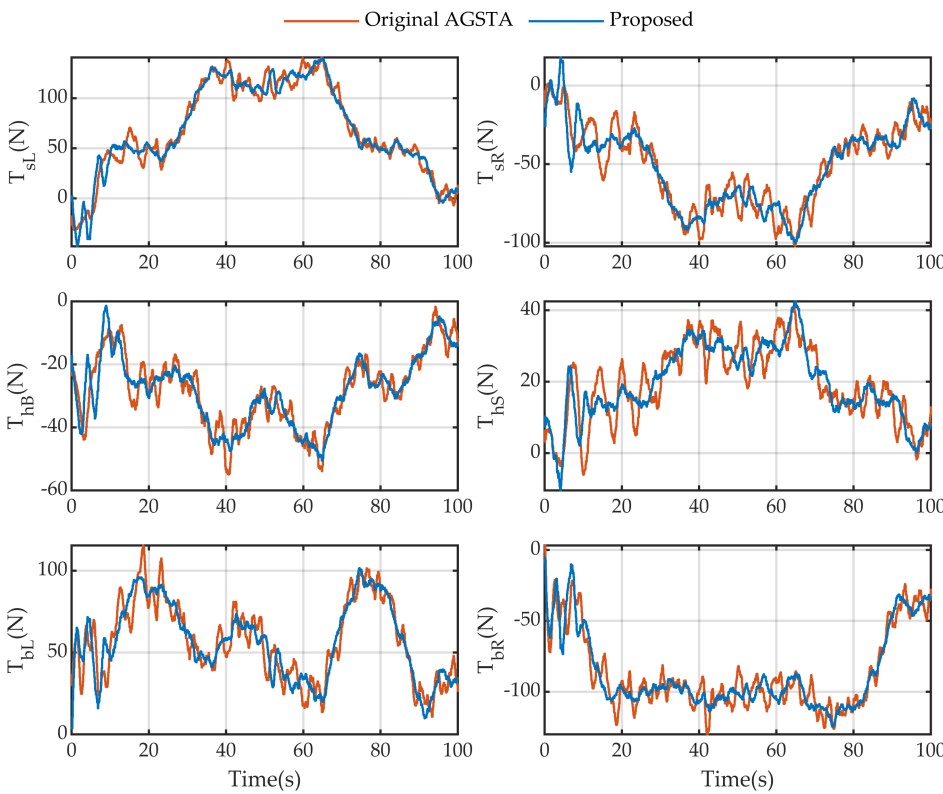

**Figure 7.** Thrust force for each thruster during station-keeping control.

It should be noted that the control parameters of the original AGSTA controller are also tuned by the multi-objective optimization method. Hence, the CHAT value gap between the two controllers is not very obvious. However, when tunning the control parameters of the original AGSTA controller manually, we find it often leads to severe chattering and even becomes unstable. As a result, the parameter tuning process becomes hard. The proposed controller avoids the problems above, making it safer and easier to tune than the original one.

Figure 8 depicts the time trajectories of the sliding variables during the station-keeping control. It can be observed that the sliding variable enters the predefined neighborhood of origin and try to remain there. The evolution of the adaptive gains $k_1(t)$ is depicted in Figure 9; it can be seen that the control gains of the proposed controller increase or decrease

in a bounded region. The control gains tend to maintain at a small value. If the sliding variable exceeds the predefined neighborhood of origin, the adaptive gains will increase until it enters this domain again. Figure 9 shows that small control gains are sufficient to resist the disturbances and ensure an ideal sliding mode most of the time. The proposed controller ensures the sliding variable converges to the neighborhood of origin without overestimating control gains.

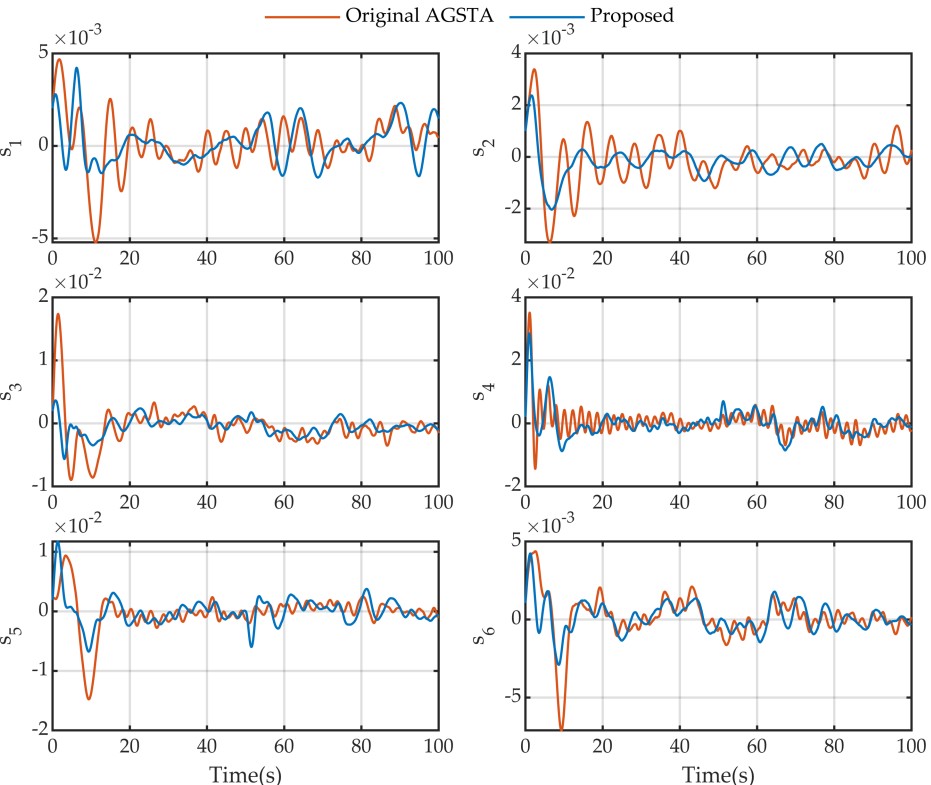

**Figure 8.** Time trajectories of sliding variables during station-keeping control.

For the AGSTA controller, the gains $k_1(t)$ keep increasing until the sliding mode is reached, then the gains remain unchanged, ensuring an ideal sliding mode for a while. As the disturbance increases, the sliding mode may be lost, so the gains $k_1(t)$ keep increasing until the sliding mode is reached again. When the disturbance decreases, the gains still maintain this value. As a result, the AGSTA may overestimate the control gains, which leads to an increase in the chattering amplitude.

It should be noted that some papers [43,49] also allow the control gains to decrease according to specific rules. However, the adaptive law proposed in these papers is limited to the conventional super-twisting algorithm, and control gains will vibrate around the set minimum value. Figure 10 compares control gains' evolution when using the traditional adaptive law and the proposed adaptive law. The magnified view of Figure 10 confirms that gains of traditional adaptive law vary in zigzag motion after arriving at the minimum value. Our proposed adaptive law avoids this problem which helps further mitigate the chattering. It should be noted that zigzag motion induced by traditional adaptive law will only appear after the controller is discretized and becomes severe with the increase of sampling time.

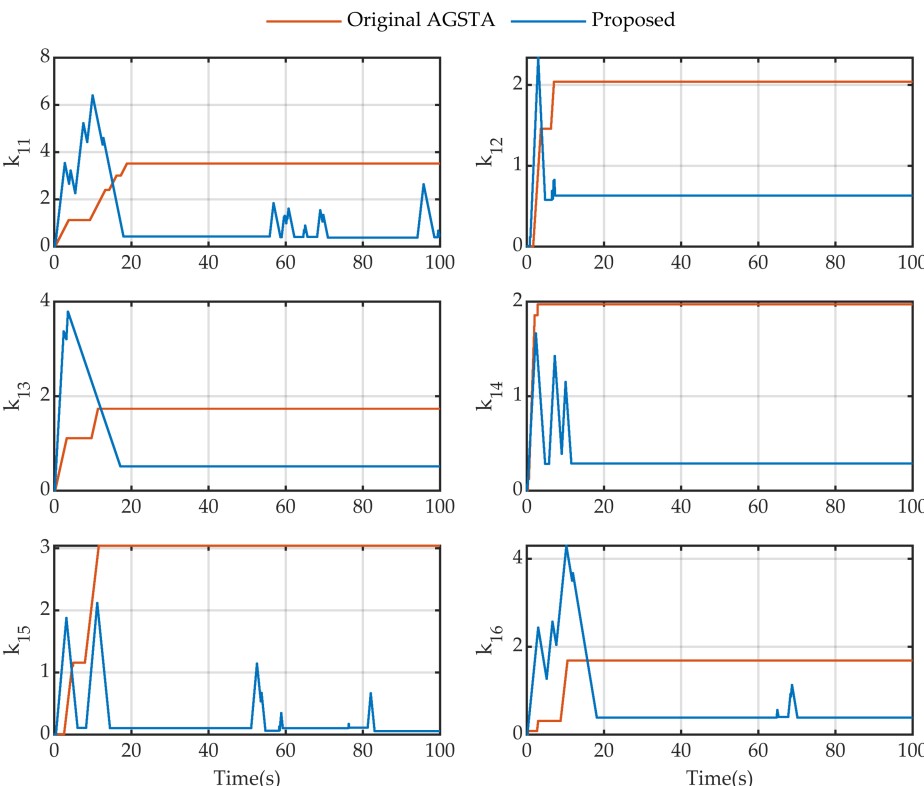

**Figure 9.** Evolution of the adaptive gains $k_1(t)$ over time.

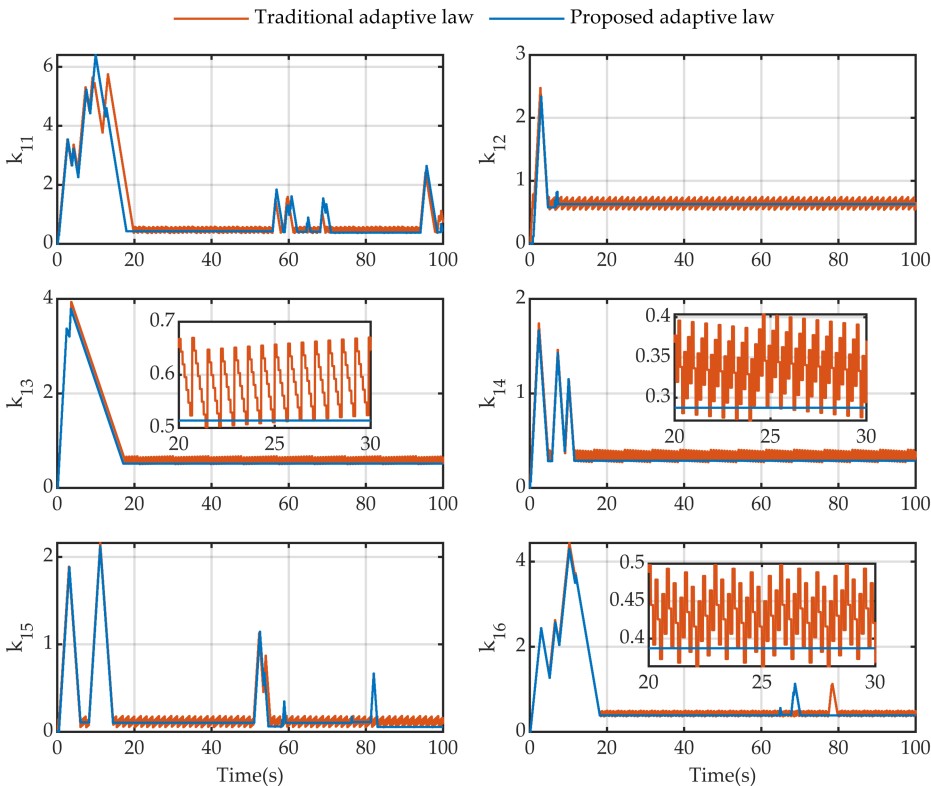

**Figure 10.** Comparison of adaptive gains between the traditional and proposed adaptive law.

The total disturbance estimated by RESO is presented in Figure 11. The results show that the total disturbance can be effectively estimated and compensated by the RESO. The initial peaking phenomenon due to the initial state estimation error is eliminated. One can

see that the estimated total disturbance has a slight phase lag compared to the simulated value because the bandwidth of RESO is restricted to the measurement noise.

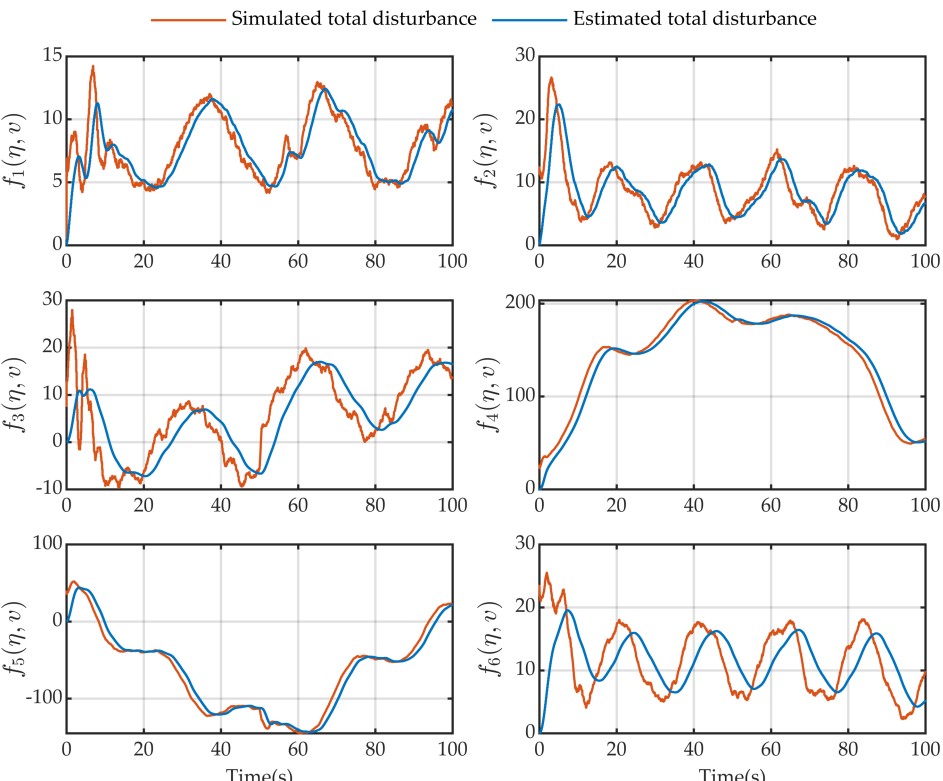

**Figure 11.** Comparison of estimated total disturbance and simulated total disturbance.

## 5. Conclusions

This paper is dedicated to solving the station-keeping control problem of ARVs for free-floating manipulation. A modified AGSTA enhanced by ATD and RESO is developed. Comparative simulation results on the Haidou-1 ARV demonstrate that the proposed control scheme is more efficient than the original AGSTA controller. The adaptive gains increase or decrease in a bounded region according to the magnitude of the sliding variable. As a result, the sliding variables are confined to a predefined neighborhood of zero without overestimating control gains, thereby reducing the chattering. Besides, the proposed new variable gain strategy completely avoids the adaptive gains varying in zigzag motion near the set minimum value and is not affected by the sampling time.

In future work, pool experiments will be conducted to demonstrate the effectiveness of our proposed controller. Besides, the proposed controller is designed in the continuous-time domain and discretized by the explicit Euler method, which will introduce discretization chattering [50]. A proper discretization scheme is necessary. Recently, the implicit discretization method has been successfully used to discretize the super-twisting algorithm [51,52], which has been shown theoretically and experimentally to provide significant chattering alleviation. It may apply to our proposed controller and further improve its chattering reduction ability. Besides, the RESO used in this paper is sensitive to noise, which should be handled in future work.

**Author Contributions:** Conceptualization, methodology, simulation, writing—original draft, N.D.; conceptualization, funding acquisition, supervision, resources, review and editing, Y.T.; visualization, validation, Z.J.; validation, review and editing, Y.B.; formal analysis, data processing, S.L. All authors have read and agreed to the published version of the manuscript.

**Funding:** This research was supported by the National Key Research and Development Program of China (Grant No. 2021YFF0306201), National Natural Science Foundation of China (Grant No.

62173320), Liaoning Province "Xingliao" Talent Program (Grant No. XLYC1807234) and the State Key Laboratory of Robotics (Grant No. 2020-Z04).

**Institutional Review Board Statement:** Not applicable.

**Informed Consent Statement:** Not applicable.

**Data Availability Statement:** Not applicable.

**Acknowledgments:** The valuable comments from the anonymous reviewers are highly appreciated.

**Conflicts of Interest:** The authors declare no conflict of interest.

## Appendix A

$$
M = \begin{bmatrix}
m - \rho X'_{\dot{u}} & -\rho X'_{\dot{v}} & -\rho X'_{\dot{w}} & 0 & mz_G - \rho X'_{\dot{q}} & -my_G - \rho X'_{\dot{r}} \\
-\rho Y'_{\dot{u}} & m - \rho Y'_{\dot{v}} & 0 & -mz_G & 0 & mx_G - \rho Y'_{\dot{r}} \\
\rho Z'_{\dot{u}} & 0 & m - \rho Z'_{\dot{w}} & my_G & -mx_G - \rho Z'_{\dot{q}} & 0 \\
0 & -mz_G - \rho K'_{\dot{v}} & my_G - \rho K'_{\dot{w}} & I_{xx} - \rho K'_{\dot{p}} & -I_{xy} & -I_{xz} \\
mz_G - \rho M'_{\dot{u}} & 0 & -mx_G - \rho M'_{\dot{w}} & -I_{yx} & I_{yy} - \rho M'_{\dot{q}} & -I_{yz} \\
-my_G - \rho N'_{\dot{u}} & mx_G - \rho N'_{\dot{v}} & 0 & -I_{zy} & -I_{zy} & I_{zz} - \rho N'_{\dot{r}}
\end{bmatrix}
$$

$$
C(v) = \begin{bmatrix}
0 & 0 & 0 & m(y_G q + z_G r) & m(w - x_G q) & -m(x_G r + v) \\
0 & 0 & 0 & -m(w + y_G p) & m(z_G r + x_G p) & m(u - y_G r) \\
0 & 0 & 0 & m(v - z_G p) & -m(u + z_G q) & m(x_G p + y_G q) \\
-m(y_G q + z_G r) & m(w + y_G p) & m(z_G p - v) & 0 & 0 & I_{yz} r + I_{xy} p - I_{yy} q \\
m(x_G q - w) & -m(z_G r + x_G p) & m(u + z_G q) & I_{yz} q + I_{xz} p - I_{zz} r & I_{yz} q + I_{xz} p - I_{zz} r & -I_{xz} r - I_{xy} q + I_{xx} p \\
m(x_G r + v) & m(y_G r - u) & -m(x_G p + y_G q) & -I_{yz} r - I_{xy} p + I_{yy} q & -I_{yz} r - I_{xy} p + I_{yy} q & 0
\end{bmatrix}
$$

$$
D(v) = -\rho \begin{bmatrix}
X'_u + X'_{u|w|}|w| & X'_v + X'_{v|v|}|v| & X'_w + X'_{\delta_s}|w|\left(\delta_s - \frac{\pi}{2}\frac{\delta_s}{|\delta_s|}\right) & 0 & X'_q & X'_r \\
Y'_u & Y'_v + Y'_{v|v|}|v| & 0 & 0 & 0 & Y'_r \\
Z'_u & 0 & Z'_w + Z'_{w|w|}|w| + Z'_{\delta_s \delta_s}|w|\left(\delta_s - \frac{\pi}{2}\frac{\delta_s}{|\delta_s|}\right)^2 & 0 & Z'_q & 0 \\
K'_{u|w|}|w| & K'_v & K'_w & K'_p & 0 & 0 \\
M'_u + M'_{uw}w & 0 & M'_w + M'_{w|w|}|w| + M'_{\delta_s}\left(\delta_s - \frac{\pi}{2}\frac{\delta_s}{|\delta_s|}\right)w + M'_{\delta_s \delta_s}\left(\delta_s - \frac{\pi}{2}\frac{\delta_s}{|\delta_s|}\right)^2|w| & 0 & M'_q & 0 \\
N'_u & N'_v + N'_{v|v|}|v| & 0 & 0 & 0 & N'_r
\end{bmatrix}
$$

$$
g(\eta) = \begin{bmatrix}
(G - B)\sin\theta \\
-(G - B)\cos\theta\sin\phi \\
-(G - B)\cos\theta\cos\phi \\
-(y_G G - y_B B)\cos\theta\cos\phi + (z_G G - z_B B)\cos\theta\sin\phi \\
(z_G G - z_B B)\sin\theta + (x_G G - x_B B)\cos\theta\cos\phi \\
-(x_G G - x_B B)\cos\theta\sin\phi - (y_G G - y_B B)\sin\theta
\end{bmatrix}
$$

**Table A1.** Dynamic parameters of the Haidou-1 ARV.

| Coefficient | Value | coefficient | Value | Coefficient | Value | Coefficient | Value |
|---|---|---|---|---|---|---|---|
| $X'_u$ | −0.601 | $X'_v$ | 0.115 | $X'_w$ | 0.031 | $X'_q$ | −0.345 |
| $X'_r$ | 0.008 | $X'_{\dot{u}}$ | −1.419 | $X'_{\dot{v}}$ | 0.051 | $X'_{\dot{w}}$ | 0.063 |
| $X'_{\dot{q}}$ | 0.165 | $X'_{\dot{r}}$ | 0.073 | $X'_{v|v|}$ | −2.564 | $X'_{u|w|}$ | −0.194 |
| $X'_{\delta_s}$ | 1.148 | $Y'_u$ | 0.044 | $Y'_v$ | −4.199 | $Y'_r$ | 0.753 |
| $Y'_{\dot{u}}$ | −0.117 | $Y'_{\dot{v}}$ | −12.006 | $Y'_{\dot{r}}$ | 1.361 | $Y'_{v|v|}$ | 7.257 |
| $Z'_u$ | −0.352 | $Z'_w$ | −0.617 | $Z'_q$ | 0.231 | $Z'_{\dot{u}}$ | −0.029 |
| $Z'_{\dot{w}}$ | −3.085 | $Z'_{\dot{q}}$ | −1.174 | $Z'_{w|w|}$ | −0.343 | $Z'_{\delta_s\delta_s}$ | −0.967 |
| $K'_v$ | −0.307 | $K'_w$ | −0.114 | $K'_p$ | 0.179 | $K'_{\dot{v}}$ | 0.048 |
| $K'_{\dot{w}}$ | 0.030 | $K'_{\dot{p}}$ | −0.347 | $K'_{u|w|}$ | −0.271 | $M'_u$ | 0.006 |
| $M'_w$ | 0.0002 | $M'_q$ | −0.774 | $M'_{\dot{u}}$ | −0.061 | $M'_{\dot{w}}$ | 0.428 |
| $M'_{\dot{q}}$ | −1.930 | $M'_{uw}$ | 1.271 | $M'_{w|w|}$ | −0.150 | $M'_{\delta_s\delta_s}$ | −1.775 |
| $M'_{\delta_s}$ | −0.051 | $N'_u$ | 0.006 | $N'_v$ | 1.318 | $N'_r$ | −2.12 |
| $N'_{\dot{u}}$ | 0.004 | $N'_{\dot{v}}$ | 2.549 | $N'_{\dot{r}}$ | −5.087 | $N'_{v|v|}$ | −0.599 |
| $m$ | 4325.925 kg | $\rho$ | 1027.77 kg/m$^3$ | $I_{xx}$ | 999 kg·m$^2$ | $I_{yy}$ | 4036 kg·m$^2$ |
| $I_{zz}$ | 3703 kg·m$^2$ | $I_{xy}$ | 0 kg·m$^2$ | $I_{yz}$ | 0 kg·m$^2$ | $I_{xz}$ | 38 kg·m$^2$ |
| $x_B$ | 0 m | $y_B$ | 0 m | $z_B$ | 0 m | $x_G$ | 0 m |
| $y_G$ | 0 m | $z_G$ | 0.05 m | $g$ | 9.8 m/s$^2$ | B | (m+2)*g |

## Appendix B

The thruster arrangement of the Haidou-1 ARV is shown in Figure 1.

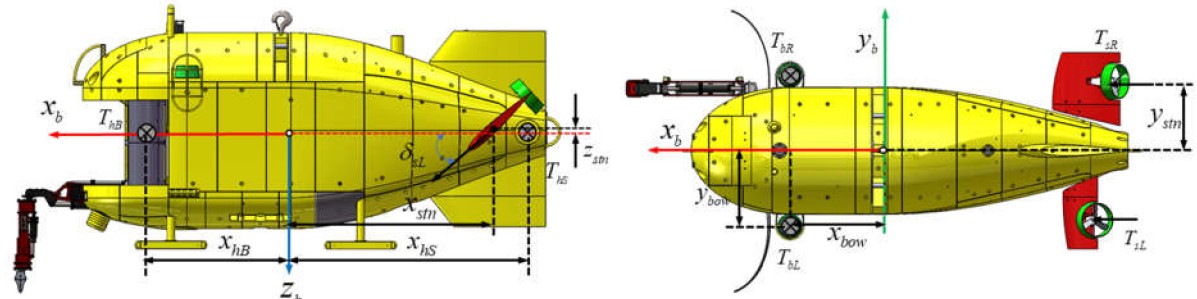

**Figure 1.** Thruster arrangement of the Haidou-1 ARV.

The thruster configuration matrix of the Haidou-1 ARV is given as:

$$
B = \begin{bmatrix}
\cos\delta_{sL} & \cos\delta_{sR} & 0 & 0 & 0 & 0 \\
0 & 0 & 1 & 1 & 0 & 0 \\
-\sin\delta_{sL} & -\sin\delta_{sR} & 0 & 0 & 1 & 1 \\
y_{stn}\sin\delta_{sL} & -y_{stn}\sin\delta_{sR} & 0 & 0 & -y_{bow} & y_{bow} \\
z_{stn}\cos\delta_{sL} + x_{stn}\sin\delta_{sL} & z_{stn}\cos\delta_{sR} + x_{stn}\sin\delta_{sR} & 0 & 0 & -x_{bow} & -x_{bow} \\
y_{stn}\cos\delta_{sL} & -y_{stn}\cos\delta_{sR} & x_{hB} & x_{hS} & 0 & 0
\end{bmatrix}
$$

where $\delta_{sL} = -72°$, $\delta_{sR} = -55°$, $x_{stn} = -1.722\,\text{m}$, $y_{stn} = 0.6$ m, $z_{stn} = -0.02$ m, $x_{bow} = 0.75$ m, $y_{bow} = 0.55$ m, $x_{hB} = 0.8$ m and $x_{hS} = -1$ m.

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
