# Peer review of "Station-Keeping Control of Autonomous and Remotely-Operated Vehicles for Free Floating Manipulation"

_jmse, doi:10.3390/jmse9111305_

Round 1

Reviewer 1 Report

The topic of the paper is within the scope of the journal and this paper is quite interesting. Overall, the paper is well written. In my opinion, it is a study to deal with a necessary problem, however, some contents need to be addressed to meet the quality publication. The work has potential for publication but, I would suggest major revision as follow:

1/ Despite the motivating topic, the novelty of the approach is rather low since it seems all the methods are existing and the experimental contribution does not consider. Thus, I am concerned about the application and practicability of this theory for the UVMS system.

2/ In the introduction, literature review is decently done but suffers from lack of recent publications. More recently published papers related to ROV with umbilical cable, UVMS systems. The concept of sliding mode control should be discussed in terms of a deeper state of the art, some new works related to sliding mode control of second-order systems, especially the dynamic sliding mode control methods, robust sliding mode method, multiple sliding mode methods, and so on, should be included. To help the authors in this direction, I suggest the following reference: https://doi.org/10.3390/s20092633, DOI: 10.6119/JMST-017-1226-18, https://doi.org/10.3390/math9161935 , DOI: 10.1109/ACCESS.2021.3098327 , https://doi.org/10.3390/s21030747 , https://doi.org/10.3390/math8081371. And the introduction should be added to do a better job of explaining the existing methods and why they are or are not valuable.

3/ In section 2.1 “Vehicle dynamics”, the dynamics of underwater vehicle should be explained more, all assumptions and physical constraints should be provided. The author can refer to the “UUV Dynamic Modeling” section of the following paper: https://doi.org/10.3390/s20051329

4/ The underwater vehicle in this paper is a fully-actuated system with 6 thrusters, the authors need to add one figure of thruster arrangement of the system and a detailed explanation of the thruster forces and moments that need to be explained. The author can refer to the “configuration of thrusters” section of the following paper: DOI: 10.1109/ACCESS.2020.3048706

5/ In section 4.1, line 414 what is a 7-functions electric manipulator? Please clarify this.

6/ In section 4.2, what is the manipulator's initial state?

7/ It is better to explain how the values of these control parameters are adjusted?

8/ The parameters for simulation such as hydrodynamic coefficients of the system need to be added in the paper. A table of all hydrodynamic coefficients of the ROV should be given in the paper.

9/ In the simulation part, simulation analysis is insufficient. More simulation results need to be added to the paper for comparisons. This paper studied the UVMS system with 6 DOF underwater vehicle and 6DOF manipulator. The reviewer thinks that the 6 angles of manipulator with respect to the time need to be performed. Also, what are the results of joint torque of manipulator? And the explanations and analysis of simulation results should be enriched to show the validity of the data.

10/ The analysis in this paper should be supported by experimental results. The authors should use practical systems to validate the proposed methods with experiment results. This paper now is difficult to prove the advantages of the proposed algorithm.

11/ Conclusion needs extended elaboration on the topic, results, lessons learned, and future works. My suggestion is to work on it from scratch to better highlight the results and possible extensions and improvements.

12/ The reference section of this paper is not properly presented and it shows some clumsiness in the paper. Refine your paper and do neat formatting of your paper. References 13, 14 are same, the same problem with references 16, 19 as well as references 33 and 49. Also, check references 25, 32, 44.

13/ Some typos and grammatical errors should be checked carefully, and some formatting problems need to be modified.

Reviewer 2 Report

The paper deals with the problem of station keeping or dynamic positioning of an ARV. The paper employs some innovative methods which potentially yields better performance than other methods. Namely, the paper proposes the adoption of AGSTA sliding mode controller and a RESO state observer to estimate the effect of unmodelled dynamics. The paper also proposes the employment of a ATD filter for the tracking reference. The paper provides a stability proof of the proposed method and simulations using a realist model of an ARV. The paper is well written and contains interesting results. There are however minor issues that should be addressed.

  • For section 3.1 to be comprehensible it is crucial to provide more information about the function fhan.
  • In equation 13 where it is \hat{X}2 it should be X2
  • Equation 15 should contain βand β2 instead of βand β1
  • In line 335 it is made a reference to an Appendix A that does not exist.
  • In line 415 it should be mentioned that D-H stands for Denavit-Hartenberg

Reviewer 3 Report

The paper presents an adaptation of methods previously used in an attempt to offer improved performance for station keeping of a subsea vehicle. The authors detail several results of a vehicle under disturbances and show the level of station keeping performance achieved when simulating the system in MATLAB, as well as improvements to other parameter outputs. 

Overall the paper is written fairly well and is sound technically, but there are some points I would make that may improve the manuscript:

  1. Some of the results figures slightly confuse me - I don't think you need a figure for both the generalized control input and the thrust force of each thruster. If anything, I would depict the thrust force in each plane of motion, as this would show any discrepencies between the two.
  2. Relating to the above comment, the authors state in the introduction that the vehicle they are modelling has poorer performing thrusters, but do not consider a thruster model for this. If this is the case would it not be sufficient to consider a dynamic model for the thrusters instead of an instantaneous model as defined in Eq. 2?
  3. On page 7 the authors state Cd = 1 as an empirical value but give no background on how this was determined. 
  4. The introduction is good overall and highlights issues with existing literature, but I would recommend synthesising between references better rather than simply listing. Also, there is a comment on page 1 that states "ARV communicates with the support vessel through an optical fiber instead of an umbilical" - if you're referring to autonomous vehicles isn't the whole point that they have no cables at all, not an optical fiber cable?
  5. The results and discussion section needs more work - there needs to be quantitative discussion about the performance, using values to justify your claims about performance.
  6. I would also be interested to see more meaningful disturbances, for example if testing in shallower waters you could simulate wave disturbances from real data. On this note, it is unclear to me whether the disturbances stated are superimposed or considered separately from one another.
  7. There is no mention of what depth the vehicle is operating at - more information should be given about the scenario you are simulating. The only mention I can see is in the vehicle state which I infer as being at 3cm deep which I don't think is correct so this needs to be clearer.

In terms of minor comments:

  1. Fig 1 should be presented on page 2 and be larger and clearer.
  2. There are quite a lot of short sentences; I would recommend using punctuation to link some of these better.
  3. Some abbreviations are given without first giving the full expanded term, for example ARV on page 1.
  4. Fig. 3 does not really add much value; I struggle to see much difference between the images except in the case of t=49s.

Overall the authors effort is noted but I would recommend alterations to be made before re-submitting.

Round 2

Reviewer 1 Report

Thank you for the revised manuscript. In a general way most of my comments were answered by the authors. The manuscript now is acceptable for publishing